# A Two-Year Clinical Description of a Patient with a Rare Type of Low-GGT Cholestasis Caused by a Novel Variant of *USP53*

**DOI:** 10.3390/genes12101618

**Published:** 2021-10-14

**Authors:** Olga Shatokhina, Natalia Semenova, Nina Demina, Elena Dadali, Alexander Polyakov, Oxana Ryzhkova

**Affiliations:** Federal State Budgetary Institution “Research Centre For Medical Genetics”, 115478 Moscow, Russia; mironovich_333@mail.ru (O.S.); semenova@med-gen.ru (N.S.); ndemina47@mail.ru (N.D.); genclinic@yandex.ru (E.D.); apol@dnalab.ru (A.P.)

**Keywords:** *USP53*, low-GGT cholestasis, novel mutation, additional clinical signs

## Abstract

Here, we report a novel truncating mutation in the ubiquitin-specific peptidase gene (*USP53*) causing low-γ-GT (GGT) cholestasis. Genetic testing was carried out, including clinical exome sequencing for the proband and Sanger sequencing for the proband and his parents. The proband harbored a novel c.1017_1057del (p.(Cys339TrpfsTer7)) mutation in the ubiquitin carboxyl-terminal hydrolase (UCH) domain of *USP53*; we describe the clinical and laboratory features of the patient with a rare type of low-GGT cholestasis caused by this variant. The clinical presentation was found to be similar to that of phenotypes described in previous studies. However, there was an unusual presence of liver hemangiomas observed in our patient. Thus, our report reinforces the link between *USP53* mutations and cholestasis. With this report, we confirm *USP53* as the gene for low-GGT cholestasis and describe liver hemangiomas as a possible additional symptom of the phenotype spectrum. The inclusion of *USP53* in the OMIM database and liver gene panels can further increase the effectiveness of molecular genetic studies.

## 1. Introduction

Neonatal physiological jaundice is a common and mostly benign symptom that typically resolves within 2 weeks after birth [1]. Nevertheless, neonatal cholestasis (NC), which is indicated by conjugated hyperbilirubinemia, is always pathologic and indicates the presence of hepatobiliary dysfunction. NC can be induced by a defect in the intrahepatic production or the transmembrane transport of bile components or a mechanical obstruction preventing bile flow [2]. The incidence of NC is approximately 1 in 2500 live births [3]. 

γ-glutamyl transferase (GGT) activity in the serum is a marker of hepatobiliary injury, especially cholestatic and biliary effect. Normal- and low-GGT cholestasis often implies hereditary hepatopathy; however, genetic variants are observed in only 75–80% of paediatric patients with this pathology [4]. 

Previous studies identified a new type of normal-GGT cholestasis caused by autosomal recessive pathogenic variants in ubiquitin-specific peptidase 53 (*USP53*). In one of the families that presented with cholestasis and hearing loss, the homozygous pathogenic variant c.951del: p.(Phe317Leufs*6) was identified in three related patients [5]. *USP53* encodes a protein of uncertain function and USP53 deficiency causes deafness in mice. Since USP53 colocalizes and interacts with the tight junction scaffolding proteins TJP1 and TJP2 in a mouse-deafness model, it has been suggested that USP53 is a component of the tight junction complex. Biallelic variants in TJP2, the human orthologue of Tjp2, cause hearing impairment and low-GGT intrahepatic cholestasis, with elongated hepatocyte–hepatocyte tight junctions. This mutation truncates the ubiquitin carboxyl-terminal hydrolase (UCH) domain, which critically regulates protein turnover by modulating deubiquitination [5]. More recently, these authors described two *USP53* homozygous variants (c.951delT; p.(Phe317Leufs*6) and c.1744C > T; p.(Arg582*) in five additional cases [6]. In another study, seven families presenting with low-GGT intrahepatic cholestasis were described. These families carried seven truncating and three missense variants. Their liver biopsy data indicated abnormalities in tight junctions, with elongation, which resembled those in TJP2 disease [7]. In another study, authors identified seven patients with homozygous mutations in *USP53*. Six of the seven patients had deletion mutations, and one had a missense mutation. Cholestasis was biochemically mild and intermittent, and responsive to medication. Liver fibrosis was present in all patients who were biopsied, and splenomegaly was apparent in five of seven patients at the last ultrasound [8]. Despite all of that, the causality is still questioned and *USP53* is not yet a valid OMIM gene.

In this study, we describe a novel truncating variant in the UCH domain of USP53, which caused the same phenotype in the patient as in the previous study. We present the description of clinical features and laboratory data of the disease in this proband.

## 2. Materials and Methods

Clinical data: The proband was examined at the Research and Counseling Department of the Research Centre for Medical Genetics (RCMG, Moscow). His initial assessment included a full history and physical examination. Laboratory studies were also conducted.

Genetic testing: Blood samples from the proband and unaffected parents were collected, and genomic DNA was extracted by standard methods. Clinical exome sequencing was performed for the proband. Target enrichment with a SeqCap EZ HyperCap Workflow solution capture array (Roche Sequencing Solutions Inc., Santa Clara, CA, USA), including the coding regions of 6640 genes currently described as clinically significant in the OMIM and The Human Gene Mutation Database (HGMD), and sequencing were carried out using Illumina NextSeq 500 (Illumina, San Diego, CA, USA). The coding sequence of *USP53* was completely covered when using this method. Sequencing data were processed using a standard computer-based algorithm from Illumina and BaseSpace software (Enrichment 3.1.0). Sequenced fragments were visualized with Integrative Genomics Viewer (IGV) software (© 2013–2018 Broad Institute, and the Regents of the University of California, CA, USA). Filtering of the variants was based on their frequency of less than 1% in gnomAD and coding region sequence effects such as missense, nonsense, coding indels, and splice sites. The variants’ clinical significance was evaluated according to the guidelines for Massive parallel sequencing (MPS) data interpretation [9,10]. Sanger sequencing was carried out to validate the exome variant in the proband and its presence in the parents. To amplify the fragment encompassing the candidate variant, custom primers were used: *USP53*_12F: CAGAGTCTGTCTTTTCATGTAACAC, *USP53*_12R: CAGAGTACTAGTATTCCTAGTTGAC.

## 3. Results

### 3.1. Clinical Evaluation

The proband was a 2-year-old male of consanguineous parents from Dagestan (Lezgins) with no significant family history. Delivery was at 39 weeks of gestation with a weight of 4000 g (> 75th percentile) and a length of 54 cm (95th percentile). The APGAR score was 7/8. The perinatal period was normal. The proband passed the neonatal screening (otoacoustic emission) and has shown no signs of hearing problems. Since birth, the child had prolonged jaundice. The level of bilirubin in the neonatal period was 253 mmol/L with a prevalence of the indirect fraction. At the age of 3 months, the boy was hospitalized because of jaundice. Clinical and laboratory studies revealed elevated levels of transaminase, which was more than four times the normal, conjugated hyperbilirubinemia, and low GGT (Table 1). Urine screening did not identify any abnormal bile acid specimen. Imaging studies did not suggest bile duct obstruction or malformation. Abdominal ultrasound examination showed hepatosplenomegaly, and diffuse changes (fibrosis) in the liver. The echocardiogram was normal. The MS/MS analysis of acylcarnitines and amino acids in plasma was normal.

He was treated with ursodeoxycholic acid, which is a naturally occurring bile acid. He continued to have itching and a high level of alkaline phosphatase with treatment; however, transaminases and bilirubin were normalized within two years. The size of the liver showed reduction and did not show hypocalcemia, thrombocytopenia, or coagulopathy.

At the age of 1 year and 8 months, ultrasound visualization revealed multiple hemangiomas of approximately 4–6 mm in diameter in the right lobe of the liver. The level of AFP was normal. At 2 years of age, the patient’s weight was 11 kg (10th percentile), height was 82 cm (< 10th percentile). His psychomotor development was normal.

Now the patient is 3 years old. The patient’s clinical signs have remained the same. MRI does not show changes in hemangiomas, transaminase levels are normal, and there are no changes in the coagulogram; the levels of cancer markers are normal. The boy still has itching, especially in the evenings.

### 3.2. Genetic Analysis

The proband was analyzed by clinical exome sequencing. The number of reads was 28,884,097. The mean target coverage depth was 64.7×. We identified 180 variants in genes that are present in the OMIM and/or HGMD databases that occur with a low frequency in the population and are not described as benign and probably benign. Comprehensive analysis was performed based on considering parental consanguinity and associating with cholestasis. The only expected deleterious variant was a novel homozygous c.1017_1057del (p.(Cys339TrpfsTer7), NM_019050.2) truncating variant in the exon 12 region of the *USP53* gene (Figure 1). This variant was neither found in the Genome Aggregation Database (gnomAD v.2.1.1) nor among the samples of 1550 Russian patients’ exomes. Sanger sequencing confirmed a homozygous state of the variant in this family where the variant was found in both parents in heterozygous state (Figure 1).

The variant was classified as “likely pathogenic” according to the guidelines for massive parallel sequencing (MPS) data interpretation (criteria: PVS1, PM2).

## 4. Discussion

Mutations in the *USP53* gene have previously been associated with a novel autosomal recessive normal and low-GGT cholestasis. To date, fifteen families harboring “pathogenic” and “likely pathogenic” variants of *USP53* have been reported. We describe a novel *USP53* mutation in a patient with low-GGT cholestasis. In this study, we not only reinforced the link between *USP53* mutations and cholestasis but also presented the description of the clinical features and laboratory data of the patient.

Ubiquitin-specific peptidase 53 belongs to the family of deubiquitinating enzymes. It is assumed that this peptidase is a component of the tight junction complex. The likely pathogenic variant p.(Cys339TrpfsTer7) truncates the UCH domain of this protein. Previously, it was shown that this domain critically regulates protein turnover by modulating the deubiquitinating process [7]. The critical importance of this domain for protein function allowed for the characterization of this variant as the cause of cholestasis development in our patient.

In this case, it is worth noting the importance of using clinical exome sequencing in comparison with gene sequencing panels. There is no information in the OMIM database that mutations in the *USP53* gene are associated with cholestasis. Therefore, *USP53* gene was not included in most of the currently known panels. Clinical exome sequencing and whole exome sequencing are advisable in cases where the molecular diagnosis is not established after gene panel sequencing.

In Table 2, we collated existing information on clinical features of previously published cases with the clinical data of our patient. The disease began in the patient at an early age. The clinical signs included jaundice in the neonatal period, splenomegaly, and fibrosis of the liver. In addition, the patient had rarer symptoms: hepatomegaly and itching. Transaminases and bilirubin were normalized after treatment using ursodeoxycholic acid; however, the alkaline phosphatase level remained high. All this corresponds to biochemically mild cholestasis phenotypes described in patients with mutations in the *USP53* gene. In previous studies, some patients had a late-onset deafness (after 9 years). Our patient, aged 3 years, had no manifestation of hearing loss. The patient’s hearing was examined by otoacoustic emission (the neonatal screening). During his life, no additional examination was made because the boy had normal speech development. In relation to the above, we cannot rule out the possible registering of hearing loss in our proband in the last few years; however, this would require further examination.

The unusual sign in the current study was the presence of liver hemangiomas detected during the examination by ultrasound visualization. Hepatic hemangiomas are benign tumors of the liver consisting of clusters of blood-filled cavities, lined by endothelial cells. The cause of hepatic hemangiomas is currently unknown [11]. It is suggested that *USP53* is a component of the tight junction complex and interacts with the tight junction scaffolding proteins TJP1 and TJP2, which regulate intercellular barriers between both epithelial and endothelial cells [6,7]. It is possible that biallelic mutations in the *USP53* gene disrupt the formation of dense connections between endothelial cells, which leads to the development of hemangiomas. A similar mechanism is observed in the development of cerebral cavernous malformations or cavernous angiomas, ultrastructural studies of which reveal poorly formed tight junctions with gaps between endothelial cells [12]. However, we cannot be sure that hepatic hemangiomas are the specific symptom of cholestasis caused by mutations in *USP53* gene because they are the most common benign liver tumor, the incidence of which ranges from 0.4% to 20% of the total population. The issue of adding liver hemangiomas to the symptom spectrum of *USP53*-related disease requires further examination.

## 5. Conclusions

In our work, we identified a patient with biallelic “likely pathogenic” variant in *USP53*, in association with low-GGT cholestasis, which extends the reports of *USP53* mutation contributing to intrahepatic cholestasis. We suggest that because of the larger number of such reported cases of this disease, *USP53*-associated cholestasis can be confirmed as a valid disease and should, therefore, be included in the liver gene panels which will further increase the effectiveness of molecular genetic studies.

## Figures and Tables

**Figure 1 genes-12-01618-f001:**
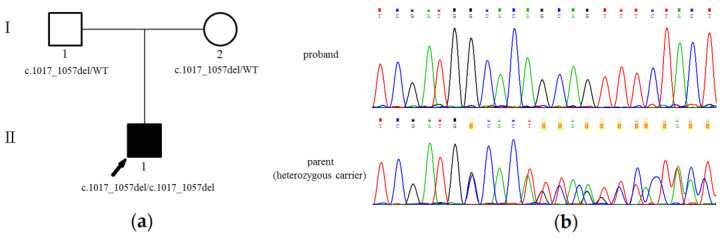
Family tree and genetic testing of the proband with a variant in *USP53*: (**a**) Family tree showing the affected proband and unaffected parents; (**b**) Sanger sequencing results demonstrating c.1017_1057del truncating *USP53* variant in the homozygous state in the proband and in the heterozygous state in both parents.

**Table 1 genes-12-01618-t001:** Blood analyses.

	**Normal**	**8 m**	**1 y 2 m**	**1 y 6 m**		**1 y 8 m**		**2 y 1 m**	
Hemoglobin	110–140 g/L	115	139	122		134		112	
red blood cells	3.5–4.5 × 10^12^/L	-	5.3	5.3		5.28		5.74	
white blood cells	6–17.5 × 10^9^/L	17.4	19.7	12.6		16.4		12	
platelets	160–390 × 10^9^/L	339	297	232		311		262	
	**Normal**	**3 m**	**4 m**	**8 m**	**11 m**	**1 y 2 m**	**1 y 6 m**	**1 y 8 m**	**2 y 1 m**
total bilirubin	5–21 µmol/L	147	245	157	22.9	38.8	26.5	33.6	6.6
direct bilirubin	< 3.4 µmol/L	124.5	120	86	14.7	18.2	16.7	15.2	2.1
ALT ^1^	0–40 U/L	185	237	359	49.5	46.4	57.9	63.3	31.6
AST ^2^	0–40 U/L	133.5	122	338	64.2	72.9		73.9	36.2
ALP ^3^	82–383 U/L	1596	1283	404	492.9	472	453.8		458.8
GGT ^4^	0–6 m: < 204; 6–12 m: < 34; 1–3 y: < 18 U/L	54	61	38.7	-	23.3	16.9	21.1	-
glucose	µmol/L	-	-	-	-	2.9	-	4.16	4.27
urea	2.8–7.2 µmol/L	-	-	3.7	3.2	5	4.1	5.1	4.3
cholesterol	3.2–5.2 µmol/L	-	4.9	3.3	-	-	2.32	2.89	3.01
total protein	64–83 g/L	-	65	65.2	-	74	-	65.6	-
albumin	35–52 g/L	-	-	37.1	-	39.7	-	39.6	-
AFP ^5^	0.5–50000 IU/ml	-	281	2459	-	5.97	-	1.95	-
calcium	2.25–2.75 µmol/L	-	-	-	-	2.59	-	2.69	-
	**Normal**	**3 m**	**8 m**	**1 y 2 m**		**1 y 6 m**	**1 y 8 m**	**2 y 1 m**	
fibrinogen	2–4 g/L	1.06	2.5	2.7		1.59	2.62	2.56	
prothrombin index	81–138%	-	52	85		96.7	90	98.3	
aPTT ^6^	25–35 sec.	-	-	35		35.6	-	32.6	

*m*—months; *y*—years; ^1^ ALT—alanine aminotransferase; ^2^ AST—aspartate aminotransferase; ^3^ ALP—alkaline phosphatase;^4^ GGT—γ-glutamyltranspeptidase; ^5^ AFP- α-fetoprotein; ^6^ aPTT—activated partial thromboplastin time.

**Table 2 genes-12-01618-t002:** Comparing clinical features of the studied patient with previously published cases.

Study	Patient	Sex	Age at Presentation	Cholestasis	Itch	Hepatomegaly	Hepatosplenomegaly	Fibrosis in the Liver	Coagulopathy	Cholangiopathy	Deafness	Additional Phenotype	Variant *USP53* (NM_019050.2)	Age of Sampling	Total Bilirubin (µmol/L)	Direct Bilirubin (µmol/L)	ALT (U/L) ^1^	AST (U/L) ^2^	GGT (U/L) ^3^	ALP (U/L) ^4^	Clinical Course (Current Age)
this study ^a^	1	M	Neonatal	L/N-GGT ^5^	+	+	+	+	-	-	-	Hepatic hemangiomas	c.1017_1057del; p.(Cys339Trpfs)	3 m	**147**	**124**	**185**	**133**	**54**	159	ANL ^8^ (3 y)
Hamoud Al Habibi et al. ^b^	1 IV:4	M	4 m	L/N-GGT	-	-	-	NK ^7^	+	-	-	-	c.951delT; p.(Phe317fs)	4 m	**172**	**158**	30	37	23	**553**	ANL (2 y)
	1a IV:5	F	15 m	L/N-GGT	-	-	-	NK	-	+	+	-	c.951delT; p.(Phe317fs)	15 m	**159**	**132**	**97**	**82**	**35**	**6316**	APLT ^9^ (24 y)
	1a IV:8	F	5 m	BRIC ^6^	-	-	-	NK	-	-	+	-	c.951delT; p.(Phe317fs)	5 m	**26**	**20**	25	29	**39**	**4432**	ANL (17 y)
	1a IV:1	M	1 y	L/N-GGT	-	-	-	NK	-	-	-	-	c.951delT; p.(Phe317fs)	1 y	**402**	**298**	**136**	**352**	30	**2557**	ANL (6 y)
	2 III:1	M	18 m	BRIC	-	+	+	+	-	-	-	Speech and developmental delay	c.951delT; p.(Phe317fs)	18 m	**155**	**85**	**51**	**61**	24	**719**	ANL (7 y)
	3 V:1	F	16 m	L/N-GGT	+	-	-	NK	-	-	-	-	c.1744C > T; p.(Arg582Ter)	16 m	**146**	**142**	36	**69**	23	**504**	ANL (1 y)
	3 IV:3	M	18 y	BRIC	-	+	+	+	-	-	-	-	c.1744C > T; p.(Arg582Ter)	18 y	**876**	**680**	**45**	**45**	**39**	**939**	ANL (35 y)
	3 IV:2	F	16 y	BRIC	-	-	-	-	-	-	-	Hypothyroidism	c.1744C > T; p.(Arg582Ter)	16 y	**179**	**128**	**63**	**80**	23	**727**	ANL (18 y)
Jing Zhang et al. ^c^	P1	F	3 d	L/N-GGT	NK	NK	NK	+	NK	NK	-	-	c.1012C > T; p.(Arg338Ter)	12 m	**23**	**19**	**184**	**215**	23	330	ANL (2 y)
	P2	M	2 d	L/N-GGT	NK	NK	NK	+	NK	NK	-	-	c.169C > T; p.(Arg57Ter) + c.831_832insAG; p.(Val279GlufsTer16)	4 m	**90**	**65**	**70**	**71**	**72**	**548**	ANL (3 y)
	P3	F	6 m	L/N-GGT	NK	NK	NK	+	NK	NK	-	-	c.569 + 2T > C + c.878G > T; p.(Gly293Val)	8 m	**212**	**159**	**103**	**121**	34	NK	ANL (5 y)
	P4	M	5 m	L/N-GGT	NK	NK	NK	+	NK	NK	-	-	c.581del; p.(Arg195GlufsTer38) + c.1012C > T; p.(Arg338Ter)	9 m	**308**	**167**	32	**84**	39	**636**	ANL (17 y)
	P5	F	1 m	L/N-GGT	NK	NK	NK	+	NK	NK	-	-	c.1012C > T; p.(Arg338Te)r + c.1426C > T; p.(Arg476Ter)	4 m	**275**	**216**	28	51	40	**543**	LF ^10^
	P6	M	5 m	L/N-GGT	NK	NK	NK	+	NK	NK	+	-	c.1558C > T; p.(Arg520Ter) + c.395A > G; p.(His132Arg)	6 m	**85**	**72**	26	41	27	342	ANL (1 y); CI ^11^ (1 y)
	P7	M	1 m	L/N-GGT	NK	NK	NK	+	NK	NK	-	-	c.297G > T; p.(Arg99Ser) + c.1012C > T; p.(Arg338Ter)	8 m	**153**	**137**	18	**225**	22	283	ANL (1 y)
Laura N Bull et al. ^d^	1	F	3 m	L/N-GGT	-	-	+	+	NK	NK	-	-	Deletion of first coding exon	3 m	**583**	**539**	**163**	NK	41	NK	ANL (11 y)
	2	M	2 m	L/N-GGT	-	-	upper limit of normal	+	NK	NK	-	-	Deletion of first coding exon	2 m	**459**	**335**	**169**	**222**	62	NK	ANL (8 y)
	3	F	5 m	L/N-GGT	-	-	+	NK	NK	NK	-	-	Deletion of first coding exon	7 m	**1626**	**1290**	**61**	**103**	46	NK	ANL (2 y)
	4	F	7 y	L/N-GGT	-	-	-	NK	NK	NK	NK	-	c.145-11_167del	NK	NK	NK	NK	NK	Normal	NK	ANL (10 y)
	5	M	Neonatal	L/N-GGT	-	-	+	+	NK	NK	NK	-	c.145-11_167del	13 y	**344**	**167**	22	38	35	NK	ANL (13 y)
	6	M	15 y	L/N-GGT	-	-	+	+	NK	NK	NK	-	c.725C > T; p.(Pro242Leu)	15 y	**300**	NK	**78**	74	25	NK	ANL (18 y)
	7	M	4 y	L/N-GGT	+	-	-	NK	NK	NK	NK	-	c.510del; p.(Ser171ArgfsTer62)	21 y	**557**	**247**	**84**	46	34	NK	ANL (21 y)

*d*—days; *m*—months; *y*—years; ^1^ ALT—alanine aminotransferase; ^2^ AST—aspartate aminotransferase; ^3^ ALP—alkaline phosphatase; ^4^ GGT—γ-glutamyltranspeptidase; ^5^ L/N-GGT—low/normal-GGT; ^6^ BRIC—benign recurrent intrahepatic cholestasis; ^7^ NK—not known; ^8^ ANL—alive with native liver; ^9^ APLT—alive, post-liver transplant; ^10^ LF—lost to follow-up; ^11^ CI—cochlear implant; Biomarker values in boldface are abnormal. *a*. Reference data: total bilirubin 5–21 µmol/L, ALT 10–40 U/L, AST 10–40 U/L, GGT 0–30 U/L, ALP 250–350 IU/L. *b.* Reference data: total bilirubin 5–17 µmol/L, direct bilirubin < 3.4 µmol/L, ALT 10–40 U/L, AST 10–40 U/L, GGT < 204 U/L, ALP 83–383 IU/L [6]. *c.* Reference data: total bilirubin 5.1–20 µmol/L, direct bilirubin 0–6 µmol/L, ALT 0–40 U/L, AST 15–60 U/L, GGT 7–50U/L, ALP 42–383 IU/L [7]. *d.* Reference data: not known [8].

## Data Availability

The data presented in this study are available in this article.

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
