# Peer review of "A Two-Year Clinical Description of a Patient with a Rare Type of Low-GGT Cholestasis Caused by a Novel Variant of USP53"

_genes, 2021, doi:10.3390/genes12101618_

Round 1
Reviewer 1 Report
The authors present a novel variant of a truncating USP53 mutation as a possible cause for low-GGT cholestasis. Whole exome sequencing has vastly expanded our knowledge on the genetic basis of cholestatic liver diseases in recent years, and many children who previously went without diagnosis have since been diagnosed and classified. To further this knowledge, case reports as the one presented by the authors are valuable and merit publication.
Nevertheless, for this particular paper I have a few concerns that should be addressed:
- The authors claim that they deliver the "most detailed description of clinical features and laboratory data". However, their "most detailed" description does not include a liver biopsy, nor does it include repeat measurements of urinary bile acids or longterm information on the development of theliver hemangiomas. While obviously it is not acceptable to perform liver biopsy in a child solely for academic reasons, the claim to be "most detailed" should maybe not be upheld, and clinical follow-up data on the hemangiomas should be added. Is any information on malabsorption of fat soluble vitamins available? Does the child still suffer from pruritus? Was any assessment of fibrosis made, e.g. transient elastography?
- Other cases of USP53 mutation associated with cholestasis are briefly mentioned in the introduction. A detailed table to collate existing information on clinical features of previously published cases would add to the value of this manuscript.
- The authors state that this patient had no manifestations of hearing loss. Was hearing formally examined, or is this clinical judgement only?
- The authors offer no possible explanation for the presence of hemangiomas. Do they have any pathophysiological speculation that might explain this finding?
Overall, I feel it is important to publish new mutations with associated clinical findings and would therefore be happy to see this case report in print, pending some improvements as detailed above.
Author Response
Dear reviewer! We eager to thank you for detailed and thorough analysis of our manuscript as well as valuable comments and recommendations to improve it. I think all the corrections will benefit the article. All changes are highlighted in blue. The corrections requested by another reviewer are highlighted in yellow. Thereunder you could find responses to each comment.
Your comments:
The authors claim that they deliver the "most detailed description of clinical features and laboratory data". However, their "most detailed" description does not include a liver biopsy, nor does it include repeat measurements of urinary bile acids or longterm information on the development of the liver hemangiomas. While obviously it is not acceptable to perform liver biopsy in a child solely for academic reasons, the claim to be "most detailed" should maybe not be upheld, and clinical follow-up data on the hemangiomas should be added. Is any information on malabsorption of fat soluble vitamins available? Does the child still suffer from pruritus? Was any assessment of fibrosis made, e.g. transient elastography?
Response:
This patient was managed by transplantologists in Moscow and doctors decided not to do a biopsy of the liver in the first year of the boy's life. After the first year, the levels of the liver's function parameters became better and we find the causing DNA changes. Moreover, the boy's family is living in the Dagestan region and can't come to Moscow because of the COVID situation. They are communicating with transplantologists by telemedicine. We spoke with the patient's parents yesterday. They told us that situation is the same. The MRI of the liver shows no changes in hemangiomas, the transaminase levels are normal, there are no changes in the coagulogram, the levels of oncomarkers were normal. The boy is continuing to have itching, especially in the evenings. Therefore, unfortunately, we don't have any reasons and possibilities for doing the liver biopsy now. In addition, the boy's parents told us that in the Dagestan they don't have the possibility to do elastography and measure the vitamins levels.
We agree with you, so we removed "most detailed" from the manuscript.
Your comments:
Other cases of USP53 mutation associated with cholestasis are briefly mentioned in the introduction. A detailed table to collate existing information on clinical features of previously published cases would add to the value of this manuscript.
Response:
We have added Table 2 to our manuscript. (Line 204)
Your comments:
The authors state that this patient had no manifestations of hearing loss. Was hearing formally examined, or is this clinical judgement only?
Response
Yes, patient's hearing was examined by otoacoustic emission. During his life, no additional examination was made, because the boy had normal speech development.
We have changed
“The proband passed the neonatal screening and has shown no signs of hearing problem.”
to
“The proband passed the neonatal screening (otoacoustic emission) and has shown no signs of hearing problem.”
We have also added an explanation in the discussion section. (Line 88-89)
Your comments:
The authors offer no possible explanation for the presence of hemangiomas. Do they have any pathophysiological speculation that might explain this finding?
Response
We have given a speculated explaination of haemangiomas in USP53 deficiency. (Line 166-181)

Reviewer 2 Report
As a new cause of neonatal cholestasis, the clinical and genetic characterization of USP53 deficiency has not been full understood yet. This paper describes a new case with novel variants.
Major: The authors describe haemangiomas in the patient. However, the authors can NOT know whether it is related to USP53 deficiency or not, and so the claim "broaden the phenotype by adding liver haemangiomas to the symptom spectrum" is too conclusive. As well, please give a speculated explaination of haemangiomas in USP53 deficiency.
Minor:
- A few more papers regarding USP53 and neonatal cholestasis have been published to the reviewer's knowledge. Please cite them prolerly.
- In line 23, "the transmembrane transport of bile". The reviewer believes "the transmembrane transport of bile components" should be more properpriate.
- Line 89, "The tandem mass spectrometry result of blood". Please indicate what are measured: amino acid spectra or any others.
Author Response
Dear reviewer! We eager to thank you for detailed and thorough analysis of our manuscript as well as valuable comments and recommendations to improve it. I think all the corrections will benefit the article. All changes are highlighted in yellow. The corrections requested by another reviewer are highlighted in blue. Thereunder you could find responses to each comment.
Your comments:
Major: The authors describe haemangiomas in the patient. However, the authors can NOT know whether it is related to USP53 deficiency or not, and so the claim "broaden the phenotype by adding liver haemangiomas to the symptom spectrum" is too conclusive. As well, please give a speculated explaination of haemangiomas in USP53 deficiency.
Response:
Thank you for the comment. We agree with the reviewer that haemangiomas in the patient are additional symptom, which hasn't described before and we are not sure that this symptom is the specific symptom of cholestasis caused by mutations in USP53 gene. We have changed
“With this report we confirm USP53 as the gene for low-GGT cholestasis and broaden the phenotype by adding liver haemangiomas to the symptom spectrum.”
to
“With this report, we confirm USP53 as the gene for low GGT cholestasis and describe liver hemangiomas as a possible additional symptom of the phenotype spectrum.” (Line 16-19)
Also, we have given a speculated explaination of haemangiomas in USP53 deficiency. (Line 166-181, this correction is highlighted in blue)
Your comments:
A few more papers regarding USP53 and neonatal cholestasis have been published to the reviewer's knowledge. Please cite them prolerly.
Response:
We have added other published articles and cited them. Thank you for noticing this. (Line 45-54)
Your comments:
In line 23, "the transmembrane transport of bile". The reviewer believes "the transmembrane transport of bile components" should be more properpriate.
Response:
In line 23: We have changed
“the transmembrane transport of bile”
to “the transmembrane transport of bile components”. (Line 26-28)
Your comments:
Line 89, "The tandem mass spectrometry result of blood". Please indicate what are measured: amino acid spectra or any others.
Response
We have changed
“The tandem mass spectrometry result of blood was normal.”
to
“MS/MS analysis of acylcarnitines and amino acids in plasma was normal.” (Line 97-98)
